# New Options for Systemic Therapies in Intrahepatic Cholangiocarcinoma (iCCA)

**DOI:** 10.3390/medicina59061174

**Published:** 2023-06-19

**Authors:** Rafał Becht, Michał P. Wasilewicz

**Affiliations:** 1Department of Clinical Oncology, Chemotherapy and Cancer Immunotherapy Pomeranian Medical University in Szczecin, 71-252 Szczecin, Poland; rafal.becht@pum.edu.pl; 2Liver Unit, Department of Gastroenterology, Pomeranian Medical University in Szczecin, 71-252 Szczecin, Poland

**Keywords:** intrahepatic cholangiocarcinoma, neoadjuvant and adjuvant chemotherapy, targeted therapies, immunotherapy, molecular diagnostics

## Abstract

Intrahepatic cholangiocarcinoma (iCCA) is a malignant neoplasm of the biliary tract, the incidence of which has increased in recent years. The etiopathogenesis is not fully elucidated, but the greatest association has been shown with inflammatory changes within the biliary tract. Surgical treatment is the main therapeutic modality; however, less than 30% of its are resectable at diagnosis, with the majority of patients requiring systemic treatment. Chemotherapy with capecitabine is the standard adjuvant therapy. For patients with inoperable tumors or metastatic lesions, chemotherapy alone or in combination with immunotherapy (durvalumab, pembrolizumab) is used. There is a need to provide systemic treatment in patients with progression after first-line treatment in good performance status. New therapeutic pathways for the treatment of this tumor type are still being identified with new emerging potential targets such as isocitrate dehydrogenase (IDH), fibroblast growth factor receptor 2 (FGFR2), or BRAF mutation.

## 1. Introduction 

Intrahepatic cholangiocarcinoma (iCCA) accounts for 10–15% of primary hepatic malignancies [1]. Histologically, it belongs to the adenocarcinomas originating from the biliary tract. The incidence of this neoplasm is gradually increasing each year. Factors that may predispose to this neoplasm are inflammatory conditions within the biliary tract, especially primary sclerosing cholangitis [2]. The median age of onset for iCCA is 70 years and the average 5-year survival is only 8% [3]. Surgery is still the most important therapeutic modality, achieving a median overall survival (OS) in the range of 27–36 months with complete R0 (excision with margins of healthy tissue) resection [4,5]. Unfortunately, only 15% of patients diagnosed with iCCA will be able to receive surgical treatment, the remaining patients being diagnosed at an inoperable stage, requiring them to receive systemic treatment [5]. The dynamic development of cancer immunotherapy and the ever-improving molecular diagnostic capabilities of tumors provide opportunities to conduct multiple clinical trials in this area. Some of those trials have already been read out and provided the basis for changing the standards of treatment in advanced iCCA tumors. The overarching goal of all systemic treatment interventions in advanced disease is prolonging the survival of patients with iCCA while ensuring a good quality of life. Figure 1 shows the treatment recommendations for intrahepatic cholangiocarcinoma (iCCA) depending on the stage and molecular characteristics of the disease.

## 2. Chemotherapy 

To improve and enhance operability in iCCA, the use of neoadjuvant chemotherapy is being investigated [3]. The decision to use this complex approach should be made by a multidisciplinary team. Not all patients will be good candidates for neoadjuvant chemotherapy. Factors such as clinical stage, the extent of infiltration, the potential for possible radical surgery as well as the patient’s performance status and organ function, including liver function, should be considered when qualifying for such a therapy option. The number of trials for neoadjuvant management is limited and efforts should be made to increase recruitment to such trials. It appears that neoadjuvant treatment may offer long-term benefits in a subgroup of patients with iCCA undergoing subsequent liver transplantation. In 2018, a paper by Lunsord et al was published, describing a series of six cases of iCCA given neoadjuvant two-drug chemotherapy (gemcitabine and capecitabine or gemcitabine and cisplatin) and subsequently referred for liver transplantation. In this small series, the observed OS at 1 year after transplantation was 100% and 83.3% at 5 years. Half of the patients relapsed at a mean time of approximately 8 months after transplantation [6]. Another single-arm, phase 2 trial currently underway for neoadjuvant treatment in iCCA is being conducted by Emory University (NCT03579771) where three drugs, gemcitabine, cisplatin and nab-paclitaxel are being used. Patients receive four courses of combination chemotherapy every three weeks followed by a hepatectomy with portal lymphadenectomy. The study is designed to determine the feasibility of this therapeutic option and to characterize adverse events associated with it [7]. Another randomized phase 2 study (NCT04506281) is investigating the use of a combination of four drugs from different therapeutic groups: gemcitabine with oxaliplatin plus lenvatinib (TKI—tyrosine kinase inhibitors) and toripalimab (PD-1antibody programmed cell death protein 1 antibody) for neoadjuvant treatment in patients with primary resectable iCCA. Patients undergo this 4-drug regimen for 9 weeks before surgery. Then, they receive eight cycles of capecitabine as postoperative treatment. The control group consists of patients directly undergoing surgery and then also receiving treatment with capecitabine postoperatively [8].

Within the group of patients who are diagnosed with an operable tumor, the radical resection of the iCCA tumor is currently supplemented with the adjuvant treatment of 6 months duration with capecitabine monotherapy. This is based on the results from the BILCAP—a randomized, controlled, multicenter phase 3 trial that compared this adjuvant treatment to observation alone. The results of the BILCAP trial showed a significant improvement in OS (53 months for capecitabine vs. 36 months in untreated patients) [9]. After subgroup analysis adjusted for prognostic factors such as lymph node status, histological grade and gender, statistical significance was not demonstrated, and the intended primary endpoint of OS was not achieved [3]. However, the BICAP trial established the clinical practice of capecitabine as an adjuvant treatment in iCCA.

In inoperable patients, the first-line use of a combination of cisplatin with gemcitabine remains the standard of care. In a randomized phase 3 ABC-02 study comparing gemcitabine plus cisplatin, vs. gemcitabine monotherapy, a subgroup of 80 iCCA patients, had a statistically significant reduction of relative risk of death by 43% (*p* < 0.01). In the same study, the median OS was 11.7 months (95% CI: 9.5–14.3) for gemcitabine with cisplatin vs. 8.1 months (95% CI: 7.1–8.7) for gemcitabine alone (hazard ratio [HR] 0.64, *p* < 0.001) [10]. In the second-line treatment of advanced iCCA, the use of 5-fluorouracil with oxaliplatin (mFOLFOX) has shown a survival benefit over best symptomatic management, as demonstrated in the phase 3 ABC-06 trial [11,12]. The multicenter randomized phase 2 NIFTY trial investigated the use of liposomal irinotecan with 5-fluorouracil and leucovorin versus 5-fluorouracil and leucovorin alone in patients with unresectable iCCA after progression on gemcitabine with cisplatin chemotherapy [13]. One hundred seventy-four (174) patients were enrolled in the study, and randomized to the liposomal irinotecan group or the control group). After a median follow-up time of 11.8 months, the median progression-free survival (PFS) was significantly longer in the liposomal irinotecan with 5-fluorouracil and leucovorin group (7.1 months, 95% CI 3.6–8.8) than in the fluorouracil and leucovorin group (1.4 months, 1.2–1.5; risk ratio 0–56, 95% CI 0.39–0.81; *p* = 0.0019). Given the data from the NIFTY study, liposomal irinotecan plus fluorouracil and leucovorin could be considered the standard of care, second-line therapy in advanced biliary tract cancer [13].

## 3. Immunotherapy 

Intrahepatic cholangiocarcinoma exhibits immunogenic features through the expression of immune checkpoint molecules, programmed cell death ligand 1 (PDL-1) and cytokine T-lymphocyte-associated protein-4 (CTLA-4) in the tumor microenvironment [14,15]. In many tumor types, chemotherapy has an immunomodulatory effect, and it has been hypothesized that the addition of immunotherapy to chemotherapy may improve outcomes in patients with intrahepatic biliary tract cancer compared with chemotherapy alone. The phase 3 TOPAZ-1 trial evaluated durvalumab with chemotherapy in patients with advanced biliary tract cancer [16]. Durvalumab is a human monoclonal antibody directed against PDL-1, which blocks the interaction of PDL-1 with PD-1 and CD80 on T cells and induces an immune response by breaking tumor evasion from immune control. In this study, patients were randomized at a 1:1 ratio to receive durvalumab with gemcitabine and cisplatin or placebo with the same chemotherapy backbone, where the combination treatment was administered for up to eight cycles of treatment and then durvalumab/placebo monotherapy was administered until disease progression or unacceptable toxicity. The study enrolled 685 patients and resulted in an estimated 24-month overall survival rate of 24.9% (95% CI, 17.9 to 32.5) for durvalumab arm and 10.4% (95% CI, 4.7 to 18.8) for the placebo. The risk ratio for progression-free survival (PFS) was 0.75 (95% CI, 0.63 to 0.89; *p* = 0.001). The objective response rate was 26.7% for durvalumab group and 18.7% for the placebo. Finally, the TOPAZ-1 result provided a groundwork for registration for the use of durvalumab with gemcitabine and cisplatin in first-line inoperable or disseminated iCCA [16]. The later KEYNOTE-966 trial was stopped after the eighth cycle but continued to be given until progression or intolerable toxicity. The KEYNOTE-966 trial enrolled 1069 patients (533 patients in the pembrolizumab group and 536 patients in the placebo group). The median survival time in the pembrolizumab group was 12.7 months (95% CI 11.5–13.6) vs. 10.9 months (95% CI 9.9–11.6) in the placebo group with a statistically significant HR of 0.83 [95% CI 0.72–0.95], *p* = 0.0034) [17]. The KEYNOTE-966 trial reaffirmed the important role of combined immunotherapy and chemotherapy in the treatment of advanced iCCA. Despite the encouraging results of TOPAZ-1 and KEYNOTE-966, it remains an open question what are the predictive factors of response to immunotherapy among patients with iCCA. It is possible, that the presence of high microsomal instability/defects in DNA mismatch repair (MSI-H/dMMR), with approximately 3% of iCCA having this disorder [18] may predict a more favorable response. The phase 2 KEYNOTE-158 study evaluated the therapeutic response to pembrolizumab in cancers other than colorectal cancer with known high microsatellite instability (MSI-H) and mismatch repair (dMMR) in patients who had previously received systemic therapy for advanced solid tumors. The KEYNOTE-158 study included 351 patients with various cancers: endometrium (22.5%), gastric cancer (14.5%), small bowel cancer (7.4%), ovarian cancer (7.1%), cholangiocarcinoma/biliary tract cancer (6.3%), pancreatic cancer (6.3%) and brain tumors (6.0%). In the cholangiocarcinoma/biliary tract cancer group, 20 patients were included in the final analysis [19]. Pembrolizumab in cholangiocarcinoma/biliary tract cancer showed clinically significant and durable benefits, with a high ORR (objective response rate) of 40.9% (95% CI 20.7–63.6%), a longer median duration of response (DOR) of 30.6 (6.2–40.5) months [19]. The median PFS (progression-free survival) was 4.2 months (2.1–24.9) and the median OS (overall survival) was 19.4 months [19]. When qualifying patients with malignant tumors for immunotherapy, biomarkers that may be predictive factors for such treatment have been sought. In 2021 Marcus and co-authors reported high mutational burden (TMB-H) in solid tumors that had received prior systemic treatment [20] as a favorable predictive marker for the use of pembrolizumab. Similar to MSI-H and dMMR, tumor mutational burden (TMB) also reflects the overall somatic genomic mutation burden within a given tumor; a higher mutational burden in turn appears to increase the likelihood of neoantigen formation and immune recognition potential [20]. A TMB score of greater than or equal to 10 mutations per megabase (10 muts/Mb) has been proposed as a threshold with a high probability of neoantigen formation and, therefore, referred to as TMB-H. Variability in response to anti-neoantigen has been observed in different types of TMB-H tumors depending on the distribution of TMB in the tumor, the clinical stage of the tumor, or factors such as the tumor microenvironment or the likelihood of neoantigen formation due to various causes of TMB (e.g., MSI, smoking) [20]. TMB-H tumors are expected to be more immunogenic and more responsive to immunotherapy. This finding was also confirmed by the authors of the prospective phase 2 study CheckMate-848 (NCT03668119). This study set out to investigate whether a PD1 inhibitor, nivolumab in monotherapy or in combination with ipilimumab is more effective in the treatment of advanced solid tumors with TMB-H [21]. More than 40 different types of solid tumors were enrolled in the study, the largest percentage of which were patients with biliary tract tumors. The results presented by the authors show that nivolumab/ipilimumab dual immunotherapy produces better responses than nivolumab monotherapy with manageable safety. The ORR of patients given dual immunotherapy was 35.3% (95% CI, 24.1–47.8%) in the cohort of patients with established TMB-High (tTMB-H) tissue disease (*n* = 68), which was higher than the ORR of 22.5% (95% CI, 13.9–33.2%) achieved by the cohort of patients with TMB-High (bTMB-H) blood disease (*n* = 80) [21]. However, the study results did not directly translate into clinical practice, where this combination treatment option is rarely considered.

## 4. Targeted Treatment

In patients who have not responded to first-line treatment with chemotherapy or chemoimmunotherapy, have progressed after such treatment, or have an unresectable or metastatic iCCA, regorafenib may be a therapeutic option. Regorafenib is an oral diphenylurea multikinase inhibitor that potently inhibits angiogenic receptors (VEGFR1-3, vascular endothelial growth factor receptor 1-3; Ang1/TIE2 - angiopoietin-1/TIE2 receptor) and stromal (PDGFR-β platelet-derived growth factor receptor beta; FGFR1, fibroblast growth factor receptor 1) factors that promote tumor neovascularization, vascular stabilization, and lymphatic vessel formation, which play a significant role in the tumor microenvironment and metastasis [22]. A phase 2 study evaluating the efficacy and safety of regorafenib was conducted, which included patients with advanced inoperable disease or metastatic biliary tract cancer after receiving one or more lines of standard treatments. A total of 43 patients were included and 62% (*n* = 27) of these were iCCA patients. A total of 34 study patients who received at least one cycle of regorafenib treatment were ultimately eligible for final analysis. The median PFS in this group was 15.6 weeks (90% CI = 12.9–24.7) and the median OS was 31.8 weeks (90% CI = 13.3–74.3) with a survival rate of 40% at 12 months and 32% at 18 months [22]. In this study, the authors concluded that regorafenib may be an effective agent in treating refractory forms of iCCA after standard first-line treatment.

Contemporary systemic treatment of advanced iCCA is rapidly moving towards personalized therapies based on molecular targets. Dynamic advances in the molecular diagnosis of cancer undoubtedly contribute to this approach. Based on DNA profiling of liquid or tissue biopsy specimens, it is estimated that up to approximately 30% of advanced iCCA have somatic lesions that can be treated with available targeted drugs [23]. KRAS and TP53 gene mutations are frequently found in large ductal iCCA arising from the perianal glands. While a subgroup of iCCA originating from the small duct glands is characterized by mutations in the isocitrate dehydrogenase (IDH) and fibroblast growth factor receptor 2 (FGFR2) gene fusions [23]. Table 1 shows possible molecular targets for therapy in iCCA.

Mutations in the isocitrate dehydrogenase (IDH) gene fusions of both IDH1/IDH2 are present in approximately 20% of patients with iCCA. One drug has successfully completed a phase 3 study ClarIDHy, which investigated ivosidenib, an oral inhibitor of the mutated IDH1 enzyme [24]. The ClarIDHy trial showed improved PFS in patients with progression after first-line treatment with ivosidenib (HR 0.37, 95% CI 0.25–0.54, *p* < 0.0001) [24]. Based on those results, the drug was initially approved by the FDA (Food and Drug Administration) in 2021 and received approval for use in Europe by the EMA (European Medicines Agency) in February 2023. 

**Table 1 medicina-59-01174-t001:** Molecular targeted therapies in intrahepatic cholangiocarcinoma (iCCA).

Molecularly Targeted Treatment iCCA		
Molecular Target	Targeted Therapy	References
VEGFR1-3, PDGFRβ, FGFR1	Regorafenib	[22]
IDH	Ivozydenib (ClarIDHy trial)	[24]
FGFR2	Pemigatinib, infigratinib, futibatinib	[25,26,27]
HER-2-mutated	Trastuzumab deruxtecan (T-DXd)	[28]
HER-2-amplified	Trastuzumab, pertuzumab	[29]
BRAF V600E	Dabrafenib (ROAR trial)	[30]
BRCA1/2, PALB2 mutation	Niraparib (NCT 03207347 trial)	[31]
NTRK	Larotrectinib, entrectinib	[32,33]
KRAS G12C mutation	Adagrasib (KRISTAL-1 trial)	[34,35]

Fibroblast growth factor (FGF) signaling plays a role in cell proliferation and angiogenesis. Fusion of the gene for fibroblast growth factor receptor 2 (FGFR2) occurs in 15–20% of iCCAs. The FGFR1-3 inhibitor pemigatinib in a phase 2 study documented clinical efficacy in patients with iCCA and FGFR2 gene fusion achieving an ORR in the range of 20–40%, a median PFS of approximately 7 months and OS of up to 17 months [25]. This formed the basis for the registration of this drug in the treatment of patients with iCCA with FGFR2 gene fusion after the second and subsequent lines of therapy. Other FGFR2 inhibitors include infigratinib and futibatinib, which have shown encouraging efficacy in iCCA in early-phase studies [26,27]. Futibatinib is currently in a phase 3 FOENIX-CCA3 trial (NCT 04093362), where it is studied as the first-line treatment of advanced iCCA patients with FGFR2 gen fusion compared to the use of gemcitabine with cisplatin [36]. However, like in most targeted agents, secondary resistance mutations to FGFR2 inhibitors have also been identified, and the re-biopsy of tissue or liquid-circulating tumor DNA should be considered to identify potential mechanisms of resistance [27].

An established predictive biomarker and promising molecular target is HER2/neu (ERBB2) which is present in 5–10% of iCCA [37]. In the phase 2 MyPathway non-randomized open-label study, which included 39 patients with metastatic cholangiocarcinoma with HER2 amplification/HER2 overexpression, double HER2 blockade was used: pertuzumab with trastuzumab [29]. The ORR was 23%, the median PFS was 4 months, and the median OS was 10.9 months. After analysis of responses among study participants, it was found that patients with HER2 amplification/HER2 overexpression without HER2 mutations had better responses compared to the poorer responding group with HER2 mutations [29]. Currently, several drugs are being introduced in early-phase clinical trials for HER2 mutations including, among others, Trastuzumab deruxtecan (T-DXd) (NCT 04482309) [28], could be a new direction of therapy for these cancers in the future.

In approximately 5% of patients with iCCA, mutations of the BRAF gene are detected, most commonly BRAF V600E. The ROAR trial, which is an open-label single-arm phase 2 study for patients with BRAF-mutated iCCA, combined a BRAF inhibitor (dabrafenib) and a MEK inhibitor (trametinib) [30]. This study demonstrated an ORR of 51% with a median PFS of 9 months and a median OS of 14 months in previously treated patients with iCCA and the BRAF V600E mutation. This is a promising therapeutic avenue for this small group of iCCA patients and requires further studies and a registration path in Europe.

Another small percentage of iCCA patients have pathogenic variants in homologous recombination DNA damage repair genes, which may be more amenable to treatment with platinum compounds and poly-(ADP-ribose) polymerase inhibitors (PARP). Patients with BRCA1/2 and PALB2 mutations who demonstrated a good response to chemotherapy with platinum (cisplatin, oxaliplatin) should be candidates for clinical trials with PARP inhibitors. A phase 2 clinical trial (NCT 03207347) investigated niraparib in BAP1-mutated tumors and DNA damage response-deficient (DDR) tumors and the study group included patients with cholangiocarcinoma [31].

NTRK (neurotrophic tyrosine receptor kinase) genes encoding TRK (tropomyosin receptor kinase) proteins can fuse with other abnormal genes, resulting in a signal for tumor proliferation. Fusions of NTRK genes are rare but occur in tumors located in different organs. In iCCA, NTRK fusions occur in <0.1% of cases. Drilon and co-authors presented the results on the efficacy of larotrectinib in adult and pediatric patients with malignancies with tropomyosin receptor kinases (TRK) fusion. The study enrolled a total of 55 patients with confirmed TRK-positive fission, who were assigned to one of three arms (protocols NCT02122913, NCT02637687 and NCT02576431) based on patient age. The primary endpoint was the overall response rate (ORR), which for the entire group was 75% (95% [CI], 61 to 85) [32]. Among the 55 patients included in this study, there were 2 patients (4%) with a diagnosis of cholangiocarcinoma with the presence of a TRK gene fusion. The authors concluded that larotrectinib showed a clear and sustained anti-tumor effect in patients with NTRK fusion-positive cancer, regardless of the patient’s age or tumor type [32]. These findings support the need to test the advanced iCCA tumors for NTRK gene fusion, which may be an option for targeted therapy in such patients. Larotrectinib was FDA approved in November 2018 was approved for patients (also of pediatric age) affected by solid tumors coupled with NTRK gene fusion. In August 2019 Entrectinib was also approved for the treatment of NTRK gene fusion tumors [33].

The KRAS (Kirsten rat sarcoma viral oncogene homolog) gene encodes a protein that is involved in the activation of a cascade of signaling pathways, including the epidermal growth factor receptor (EGFR) signaling pathway, which is thought to be fundamental in the regulation of epithelial cell proliferation, growth, and neoplastic transformation [38]. The RAS (Rat sarcoma viral oncogene homolog) protein functions as a signal transducer from activated EGFR. Activation of EGFR (by binding to its ligand) leads to activation of the RAS/RAF/MAPK and PI3K/AKT signaling pathways, followed by increased proliferation and inhibition of apoptosis in tumor cells [39]. KRAS mutations are common in iCCA patients and were identified in 20–25% of tumor cases [34]. Salem and co-authors presented at ESMO 2021 World Congress on Gastrointestinal Cancers, data on the frequency of KRAS G12C mutations in gastrointestinal cancers. The authors showed that the prevalence of the G12C mutation in gastrointestinal cancers was 4.3%. The mutation was most common in appendix cancer and colorectal cancer. Among the 1481 cases of biliary tract cancer analyzed, the KRAS G12C mutation was found in 18 (1.2%) cases [40]. The tyrosine kinase inhibitor (TKI) sotorasib, which specifically targets the KRAS G12C mutation, has been approved for the treatment of patients with adenocarcinoma of the lung [41]. The KRASG12C mutation accounts for approximately 2.3% of biliary malignancies in the Chinese population as shown by Loong et al in their paper [34]. Another KRAS G12C inhibitor, Adagrasib, in the KRISTAL-1 study, showed a good ORR of 41% among 27 KRAS G12C mutation patients with cancers in the gastrointestinal tract. In the KRISTAL-1 trial, eight patients had biliary tract cancer [35]. Several other inhibitors for KRAS G12C are currently in early-phase clinical trials and the results in patients with iCCA are awaited.

## 5. Conclusions

Intrahepatic cholangiocarcinoma (iCCA) is the second most common malignant tumor of the liver [2]. Unfortunately, the results of treatment of these tumors are not satisfactory due to the advanced stages of the disease at the time of diagnosis in most patients. Surgical treatment remains the mainstay of treatment, but only a few patients are eligible. Neoadjuvant chemotherapy appears to be an important treatment option, which can improve surgical outcomes, and increase eligibility for surgery [42]. Adjuvant chemotherapy has found an important place in the treatment of iCCA. Patients with inoperable tumors or metastatic disease have a poor prognosis. Therefore, this group of patients should undergo additional tumor testing in order to consider extended treatment options. Each patient with iCCA should have a treatment plan established based on a multidisciplinary team. The search for predictive factors in the form of molecular targets seems to be an important emerging component of the plan. As of today, iCCA tumors are one of the cancers where personalized oncology is starting to play, or is already playing an important role. Targeted therapy Pemigatinib has been approved for 10–15% of patients with FGFR2 fusions/rearrangements and Ivosidenib is approved for a subgroup with the presence of IDH1 mutations [24,43]. Immunotherapy in combination with chemotherapy for the treatment of these cancers is also very promising in improving the survival of patients with inoperable disease, as evidenced by the results of the TOPAZ-1 trial with the addition of durvalumab to chemotherapy with gemcitabine and cisplatin and KEYNOTE-966-pembrolizumab with gemcitabine and cisplatin [15,17]. Intrahepatic cholangiocarcinoma (iCCA) still requires intense research activities to identify new and effective treatments. In most cases, those efforts should be directed towards personalized medicine, as the treatment of a patient with advanced iCCA is expected to be individualized. It should be taken into account that new systemic treatments of the advanced and inoperable form of iCCA, in addition to prolonging survival, should also be evaluated in the context of maintaining a good quality of life for the patient.

## Figures and Tables

**Figure 1 medicina-59-01174-f001:**
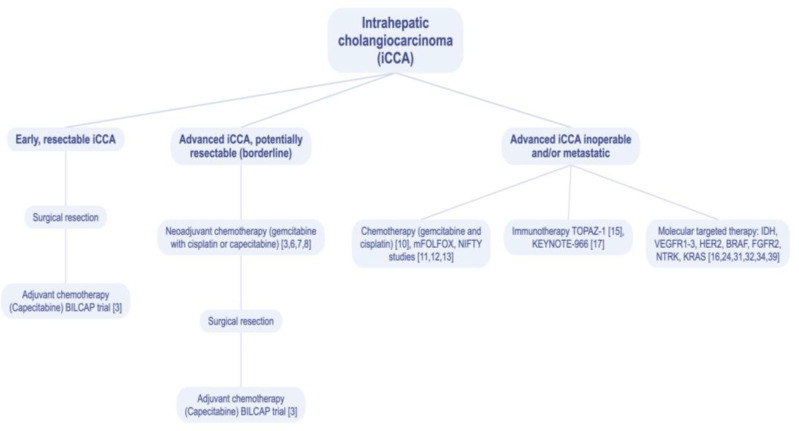
Treatment recommendations for intrahepatic cholangiocarcinoma (iCCA). VEGFR1-3—vascular endothelial growth factor receptor 1–3; HER2—human epidermal growth factor receptor 2; BRAF—type B rapid accelerated fibrosarcoma gene; FGFR2—fibroblast growth factor receptor 2; NTRK—neurotrophic tyrosine receptor kinase; KRAS—Kirsten rat sarcoma virus gene.

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
