# Peer review of "New Options for Systemic Therapies in Intrahepatic Cholangiocarcinoma (iCCA)"

_medicina, 2023, doi:10.3390/medicina59061174_

Round 1
Reviewer 1 Report
The authors reviewed the treatment options for intrahepatic cholangiocarcinoma. While the information is interesting, but the manuscript is not well organized and is not easy to follow. It also lacks graphical abstractions of the available drugs and their classification for better understanding. Also, illustrating the involved pathways and the place of drugs in those pathways is necessary to recognize this manuscript as a review, and not as a report.
The English needs revisions. Also, the sentences should be revised for a better scientific sound.
Author Response
Manuscript ID:medicina-2382801
Cover letter in response to the reviewer1.
Thank you for your valuable comments in your review of our manuscript: "New options for systemic therapies in intrahepatic cholangiocarcinoma (iCCA)". Following your comments, we have significantly changed the layout of the manuscript by dividing the text into an introduction, a section on chemotherapy, immunotherapy and targeted treatment, and a discussion at the end. We added a figure and a summary table in targeted therapy to better illustrate the theme of the manuscript. We would like to note that the manuscript focuses primarily on the practical - clinical side of systemic therapies used in iCCA. We also thank you for your comments on the English language, we have made corrections in this direction. We hope that the text presented after revisions will be acceptable for acceptance for publication at this time and will be a practical review for clinicians.

Reviewer 2 Report
This review manuscript has summarized the systemic therapies of intrahepatic cholangiocarcinoma from 1st line to late line including ongoing studies. This comprehensive review also states the strategies based on precision oncology, which advances the field of this type of cancer. However, there needs to be improved and I have some suggestions.
l The authors should include the figure to describe the molecular spectrum based on the tumor location. Biliary cancer has different molecular profiling depending on the location.
l They need to reconsider the manuscript structure. There are only Introduction and Conclusion, making it harder for the readers to understand.
l Regarding immunotherapy, new evidence is coming up. They need to include KEYNOTE-966 which has been published in Lancet very recently.
l KRAS section- they need to show the population of G12C mutation in biliary cancer, especially in iCCA. There is only one example of the Chinese population in the current manuscript. Some databases such as COSIMC or cBioportal are available.
l There are some typos- lenvatinib (Line 53), ivosidenib (Line 154).
l There are several wrong spaces in abstract and first paragraph.
Author Response
Manuscript ID:medicina-2382801
Cover letter in response to reviewer2.
Thank you for your valuable comments in your review of our manuscript : "New options for systemic therapies in intrahepatic cholangiocarcinoma (iCCA)".
In accordance with your comments, we have improved the structure of the manuscript by dividing the text into sections starting with a brief introduction , then we have identified the main directions for systemic treatment of iCCA (chemotherapy, immunotherapy and targeted therapy), the paper is concluded with a conclusion.
We added a figure and a summary table in targeted therapy to better illustrate the theme of the manuscript and corrected the typos noted.
Regarding immunotherapy, we have added extensive information on the recently published results of the KEYNOTE-966 study which provides additional information on the topic of immunotherapy.
In the chapter on targeted treatment, we analyzed the iCCA population and the presence of KRAS G12C mutation in a non-Chinese population following reports by Salem and co-authors who presented their paper at the ESMO 2021 World Congress on Gastrointestinal Cancers.
We would like to note that the manuscript focuses primarily on the practical - clinical side of systemic therapies used in iCCA.
We also thank you for your comments on the English language, we have made corrections in this direction.
We hope that the text presented after revisions will be acceptable for acceptance for publication at this time and will be a practical review for clinicians.

Reviewer 3 Report
In this review, the authors attempted to summarize the most relevant and recent data about systemic therapies (ranging from chemotherapy to immunotherapy, until to personalized therapies based on molecular targets) adopted in the treatment of intrahepatic cholangiocarcinoma (iCCA). The manuscript, in many of its parts, is written in a too much concise and very unclear way to be understandable to the reader. In particular, many of the results concerning the clinical trials cited in the text are not properly reported/explained. Both Major and minor revisions are required for publication in “Medicina”.
Major revisions:
In general, the whole part in which the authors mention and describe the clinical trials is written in a confusing and unclear way for the reader. This part should be revised with more care, giving some more information to the reader, in order to make the text easier to understand. For example:
_Lines from 99 to 106: Please the authors specify in the text if the regorafenib phase II study was performed on iCCA affected patients (are they the 43 patients mentioned in the text?), as “advanced or inoperable disease or metastatic disease” is inaccurate. Moreover, authors speak about 43 patients, “34 of whom received at least one cycle of treatment”. Did the other 9 patients receive a placebo?
_Lines from 121 to 128: The authors describe the phase II KEYNOTE study very unclearly. First, they write that “The phase 2 KEYNOTE study enrolled 22 patients with ICC and MSI-H/dMMR”, then they claim “In solid tumors with high tumor mutational burden (TMB-H), a response to immunotherapy is observed, but no iCCA patient with TMB-H was included in the study with pembrolizumab”. Written in this way, the text causes confusion in the reader. Please, the authors try to describe the results of this trial more organically, shedding light in particular on the data concerning iCCA patients, if they enrolled in the study, as intrahepatic cholangiocarcinoma is the focus of the review.
_Line 177: Authors cite in the text “HER2-amplified tumors compared with HER2-mutated iCCA”, without specifying what does mean HER2-amplified tumors in respect to HER2-mutated iCCA. Please, rewrite the sentence in order to better explain the concept.
_Line 201: The authors cite “three trials of larotrectinib treatment”, but then they list the results speaking about “this trial”. Please the authors rewrite the statement if they are referring to three trials, specifying what each of them consists of.
Minor revisions:
_Line 19: Please the authors replace “which identify new potential targets” with “which has identified new potential targets”.
_Line 31: Please the authors explicit the acronym “R0” in the text, when first mentioned.
_Line 31: Please the authors standardize in the text: either always “per cent” or always “%”.
_Line 42: The authors probably meant gemcitabine in “capecitabine or gemctabine”. Please correct the text.
_Line 43: Please the authors explicit the acronym “SO” (overall survival) in the text, when first mentioned.
_Line 48: Please the authors replace “Patients receive” with “Patients received”.
_Lines 54-55: Please the authors modify the statement “Treatment lasts nine weeks before surgery and then all receive eight cycles of capecitabine as post-operative treatment” as follow “Patients undergo this treatment for nine weeks before surgery. Then, they receive eight cycles of capecitabine as post-operative treatment”.
_Line 78: please the authors modify “with chemotherapy alone” with “with chemotherapy treated patients alone”.
_Line 79: Please the authors standardize in the text: either always “Phase 3” or always “Phase III”.
_Line 88: Please the authors explicit the acronym “CI” (confidence interval) in the text, when first mentioned.
_Line 99: The same consideration made for Line 79 is valid.
_Line 102: Please the authors explicit the acronym “PFS” (Progression-free Survival) in the text, when first mentioned.
_Line 104: Please the authors substitute “The study authors concluded” with “In this study the authors concluded”.
_Line 115: When the authors write “According to the authors of the paper…” they are referred to Ref 17 in the text? Please move the ref to the end of this sentence.
_Line 122: please the authors correct “ICC” with “iCCA”
_Line 125: Please the authors explicit the acronym “ORR” (objective response rate) in the text, when first mentioned.
_Line 130: The same consideration made for Line 79 is valid.
_Line 146: Please the authors standardize in the text: either always “per cent” or always “%”.
_Line 154: The same consideration made for Line 79 is valid.
_Line 156: Please the authors explicit the acronym “HR” (Hazard Ratio) in the text, when first mentioned.
_Line 161: The same consideration made for Line 79 is valid.
_Line 167: The same consideration made for Line 79 is valid.
_Lines from 167 to 170: please the authors replace “Futibatinib is currently in a phase 3 trial, where it is used in the first-line treatment of advanced iCCA in patients with FGFR2 gene fusion compared to the use of gemcitabine with cisplatin - the FOENIX-CCA3 trial (NCT 169 04093362)” with “Futibatinib is currently administered in the phase III FOENIX-CCA3 trial (NCT 169 04093362), where it is used in the first-line treatment of advanced iCCA patients with FGFR2 gene fusion compared to the use of gemcitabine with cisplatin”.
_Line 183: The same consideration made for Line 79 is valid.
_Line 191: Please the authors replace “who have had a good response” with “who demonstrated a good response”.
_Line 193: The same consideration made for Line 79 is valid.
_Line: Please the authors specify the acronyms NTRK and TRK (Neurotrophic Tyrosine Receptor Kinase and Tyrosine Receptor Kinase, respectively) in the text.
_Line 198: Please the authors delete “molecularly” from the statement.
_Line 198: The same consideration made for Line 146 is valid.
_Line 200: Please the authors modify “Two of 55 patients with different cancers with TRKL fusions” in “Two of 55 patients with different cancers associated to TRKL fusions”.
_Lines from 202 to 205: Please the authors rewrite as follow: “Larotrectinib was FDA approved in November 2018 for patients (also in pediatric age) affected by solid tumors coupled with NTRK gene fusion. In August 2019 also Entrectinib was approved for the treatment of NTRK gene fusion tumors”.
_Line 206: Please the authors specify the acronyms “KRAS” (Kirsten Rat Sarcoma) in the text, when first mentioned.
_Line 212: Please the authors replace “of” with “in” in “apoptosis of tumor cell”.
_Line 212-213: Please the authors rewrite as follow: “KRAS mutations are common in iCCA affected patients, as they were identified in 20-25% of tumor cases”.
_Line 213: please the authors correct “ICC” with “iCCA”.
_line 216: The same consideration made for Line 146 is valid.
Please, the authors be careful for the tenses used in the text (present, past), in order to make the text organic.
Author Response
Manuscript ID:medicina-2382801
Cover letter in response to reviewer3.
Thank you for your comments in the review of our manuscript: "New options for systemic therapies in intrahepatic cholangiocarcinoma (iCCA)". We believe they are valid and have complied with the comments and revised our manuscript. As suggested by the reviewer, we have edited the English in our paper. In the manuscript in the revised version, we have separated the thematic sections: chemotherapy, immunotherapy and targeted therapy, which will make it easier for the potential reader to select a sub-topic of interest regarding iCCA systemic therapy. We have also added a figure depicting the different routes of iCCA pajnet therapy and included a summary table on targeted therapy with references to the references so that the source material can be more easily accessed. Below we respond to your larger comments on our manuscript.
In general, in accordance with your comments, we have tried to treat more extensively the topic of the clinical trials we present and cite.
Regarding lines 99-106 "regorafenib" : A phase 2 trial of regorafenib as a single agent in patients with chemotherapy-refractory, advanced inoperable and metastatic biliary tract adenocarcinoma. Among the 43 patients included in the trial who received at least 1 dose (of regorafenib), 27 patients (62%) were with iCCA. In this study, 43 patients received at least 1 dose of the drug and 34 patients received at least 1 cycle of full treatment with the drug. Finally, a group of 34 patients who received at least 1 full course of treatment were evaluated. We have included the results and conclusions of this study in the text of our manuscript and made appropriate corrections and clarifications to make it clearer.
Regarding lines 121-128 "KEYNOTE-158" we thank you for your rightful comment, our edited text was unclear and incomprehensible, we have described the cited KEYNOTE-158 study in more detail, focusing primarily on the results regarding iCCA. Also, we clarified potential predictors of response to immunotherapy in biliary tract tumors focusing not only on MSI-H /dMMR but also touched on high tumor mutational burden (TMB-H) - which is discussed in the revised manuscript. Thank you for returning this comment to us, it is a very important insight that impinges on the proper clinical qualification of patients for immunotherapy.
Regarding verse 177 " HER2-amblified tumors / HER2-mutated tumors" Genetic alterations in HER2 are common in solid tumors and are divided into two groups: the first is hyperaddiction by gene amplification (often in breast and colorectal cancer) and the second group is activating mutations mainly affecting the HER2 kinase region (found in other lytic cancers, lung cancer but also in cholangiocarcinoma). HER2 mutated cholangiocarcinoma tumors are rare and are a clinical problem showing poor overall survival. Thank you for your valuable comment to our manuscript on this point.
Regarding line 201 "Larotrectinib" In the study cited by znas, which was dedicated to adult and pediatric patients, three arms were included (protocols NCT02122913, NCT02637687 and NCT02576431). We have corrected and described this in the recap text to make it clearer. Thank you also for this comment.
With the reviewer's indications, minor changes were made in the text, the tenses used in the text were corrected, the notation of study phases was standardized, and acronyms were clarified if they were first used in the text. All of these reviewer's comments have improved the quality of our manuscript and we hope currently will allow acceptance for publication in "Medicina".

Round 2
Reviewer 1 Report
The manuscript was much improved, but still needs improvement before publication.
Specifically, the Figure 1 needs major revision both regarding the design and also the content. It is so primitive in this format.
English quality is fine but needs proof reading.
Author Response
Manuscript ID:medicina-2382801
Thank you very much for your valuable comments in your review of our manuscript: "New options for systemic therapies in intrahepatic cholangiocarcinoma (iCCA)". Following your suggestion, we have again made language corrections and asked for help our fellow-collegue - oncologist - working at the Department of Oncology in Philadelfia (US) who is fluent in English. We also revised the figure included in the text to serve as a quick signpost for practicing clinicians in navigating through the current possible systemic therapies in iCCA.
Best Regards,
RB & MW
Reviewer 2 Report
The manuscript has been improved significantly.
The number of Conclusion part should be corrected to 5.
Author Response
Manuscript ID:medicina-2382801
Thank you very much for your valuable comments in your review of our manuscript: "New options for systemic therapies in intrahepatic cholangiocarcinoma (iCCA)". Thank you for guiding us through the revision process and for the final evaluation of our work.
Best Regards,
RB & MW
Reviewer 3 Report
The manuscript was strongly improved after corrections made on the basis of the suggestions (minor and major revisions) sent to the authors. I really appreciated the division of the text into three thematic sections (chemotherapy, immunotherapy and targeted therapy) and the addition of the figure 1 and table 1 that summaryze the current treatment regimen in iCCA affected patients and the currently in use molecular targeted therapies. In this form, I suggest the manuscript for publication in Medicina.
Author Response
Manuscript ID:medicina-2382801
Thank you very much for your comments in the review of our manuscript: "New options for systemic therapies in intrahepatic cholangiocarcinoma (iCCA)". Thank you for guiding us through the revision process and for the final evaluation of our work.
Best Regards,
RB & MW